# An Integrated Strategy for Analyzing the Complete Complex Integrated Structure of Maize MON810 and Identification of an SNP in External Insertion Sequences

**DOI:** 10.3390/plants13162276

**Published:** 2024-08-15

**Authors:** Chunmeng Huang, Yongjun Zhang, Huilin Yu, Xiuping Chen, Jiajian Xie

**Affiliations:** 1Institute of Plant Protection, Chinese Academy of Agricultural Sciences, Beijing 100193, China; cmhuang0812@163.com (C.H.); yjzhang@ippcaas.cn (Y.Z.); hlyu@ippcaas.cn (H.Y.); xpchen@ippcaas.cn (X.C.); 2Key Laboratory of Safety Assessment of Genetically Modified Organisms (Environment), Ministry of Agriculture and Rural Affairs, Beijing 100193, China; 3Plant Ecological Environment Safety Inspection and Testing Center of Ministry of Agriculture and Rural Affairs (Beijing), Beijing 100193, China; 4State Key Laboratory for Biology of Plant Diseases and Insect Pests, Institute of Plant Protection, Chinese Academy of Agricultural Sciences, Beijing 100193, China

**Keywords:** MON810, Pacbio-Hifi, single-nucleotide polymorphism, allele-specific PCR, blocker displacement amplification

## Abstract

Genetically modified maize (*Zea mays* L.) MON810 was approved for importation into China for feed use in 2004; however, the localization data concerning exogenous insertion sequences, which confer insect resistance, have been questionable. MON810 maize plants discovered in northeastern China were used to analyze the molecular characteristics of the exogenous insertion. Using PacBio-HiFi sequencing and PCR assays, we found the insertion was located in chromosome 8, and there was a *CaMV35S* promoter, *hsp70* intron, and insecticide gene *cry1Ab*, except for genome sequence insertion in the MON810 event. Importantly, the 5′ and 3′ flanking sequences were located in the region of 55869747–55879326 and 68416240–68419152 on chromosome 5, respectively. The results of this study correct previous results on the genomic localization of the insertion structure for the MON810 event. We also found a single-nucleotide polymorphism (SNP) in the *hsp70* intron, which is most likely the first SNP found in a transgenic insertion sequence. PCR amplification in conjunction with Sanger sequencing, allele-specific PCR (AS-PCR), and blocker displacement amplification (BDA) assays were all effective at detecting the base variance. The integrated strategy used in this study can serve as a model for other cases when facing similar challenges involving partially characterized genetic modification events or SNPs.

## 1. Introduction

Corn pests reduce corn production by ~3–10% and cause billions of dollars in losses each year [1]. Corn developed through transgenic technology has resulted in excellent insect resistance [2]. Transgenic corn MON810 was developed by the Monsanto Company with the gene gun transformation method, in which *cry1Ab* from *Bacillus thuringiensis* was transferred into the corn genome. Developed with anti-insect characteristics, MON810 has resistance against the European corn borer (ECB, *Ostrinia furnacalis*), Lepidoptera pests such as the Southwest Corn borer (SWCB, *Diatraea grandiosella*), and the Asian Corn Borer (ACB, *Ostrinia furnacalis*). Rigorous biosafety evaluation is necessary before genetically engineered products enter the market. Molecular biology experiments confirmed that the insertion of MON810 contained *cry1Ab* with a 3′ terminal deletion [3]. The MON810 genome contains the integrated DNA sequence in a 5.5-kb NdeI fragment containing the E35S promoter (299 bp), the maize *hsp70* intron (804 bp), and the *cry1Ab* coding region (2448 bp). The insertion sequence does not contain terminators, but a translation termination signal appears in the 9-bp region downstream of the *cry1Ab* fragment. Cutting at the ends of the insertion sequence did not affect the insecticidal activity of the expressed protein [4].

Multiple PCR-based methods can be used to characterize and identify genetic modification events. These methods can be broadly categorized as (1) element-specific, (2) construct-specific, and (3) event-specific methods [5,6,7]. Element-specific methods are mainly used for screening to determine whether a sample contains genetically modified material and to provide clues about the genetic modification event by revealing which elements are present. Construct-specific methods target the junction(s) of two or more genetically modified elements that occur together in a genetically modified organism (GMO).

Researchers have used multiple approaches to analyze the exogenous insertion structure of MON810. Thermal asymmetric interlaced PCR (TAIL-PCR), Gateway (GW), and long-distance PCR (LD-PCR) were used to obtain the sequence structure of the insertion site and surrounding genomic region, which totaled ~5.3 kb [8,9]. The contig sequence of 14,016 bp was obtained by sequence capture and third-generation sequencing, and the foreign sequence was inserted at position 55879326 on chromosome 5 according to the previous report [8]. We note, however, that the surrounding genomic sequence is discontinuous with respect to the maize genome. Thus, one point remains to be clarified about the structure. The positions of the side sequences of 5′ and 3′ were far away, so it was obvious that the current insertion sequence information was missing. Therefore, the current flanking sequence was not obtained completely and confirmed boundary position information, suggesting that the side sequence may extend to a further position. Most of the current methods for developing transgenic plants incorporate foreign DNA into the plant genome, whether it is an *Agrobacterium*-mediated transformation or direct DNA transfer. Thus, the transgene–genome boundary region will be unique for each specific transformation event. Fortunately, there is a relatively accurate sequencing technique for obtaining long sequences—PacBio-Hifi which can avoid obtaining incorrect chromosomal positioning information by obtaining short sequence lengths [10]. The PacBio long-read long-sequencing system can accurately determine the correct sequence of the sample DNA by cross-referencing the copies of each molecule, which is known as cyclic consistency sequencing (CCS) [11].

At present, the detection technology for transgenic is mainly based on four levels, but the target is a sequence, and new methods still need to be developed for the detection of a few base changes, such as SNP. Direct Sanger sequencing using PCR products can obtain accurate sequence and peak map information within a few hours, which is a simple method to analyze molecular characteristics. Allele-specific PCR is a method based on the region of variation and can detect individual samples with content as low as 10%. In contrast, the new blocker displacement amplification (BDA) technology has a wider range of applications [12]. Sample enrichment to elevate the allele fractions of rare variants can allow economical sequencing-based rare variant profiling but has been difficult to realize in highly multiplexed settings. Past demonstrations of DNA-variant enrichment employ either selective depletion of wild-type sequences via hybridization BDA [13,14,15] or selective PCR amplification of variants BDA [16]. It has been challenging to scale these approaches to multiplexed enrichment of many different variants across different loci.

Here, we present a simple method for enriching dozens of bases in a row of potential DNA sequence variants from genomic loci simultaneously. The DNA oligonucleotides used are unmodified and broadly available. The key enabling innovation was a rationally designed competitive hybridization reaction that allows PCR not only to sensitively recognize and selectively amplify even single-nucleotide variants at allele frequencies of 0.1% but also to do so across a temperature window spanning 8 °C. Our variant allele enrichment method, blocker displacement amplification (BDA), substantially reduced both the cost and the complexity of profiling rare DNA variants, making genomics analyses more accessible and economical.

In this study, the full-length sequence of MON810 was determined and verified, overcoming problems associated with the short read lengths of first- and second-generation sequencing, inaccurate chromosome localization, and incompatibility of 5′ and 3′ peripheral sequences. With this approach, we were able to obtain the full-length insertion sequence of MON810 and sort out the order of its components and sequence fragments. The results were further verified by long PCR amplification and first-generation sequencing. The PCR-Free resequencing results validate both the insertion sequence and the SNP found by chance in the long-read resequencing data. Different levels of specific detection methods established for this SNP site provided technical support for tracing the occurrence and development of variation sites. In total, our study validated diagnostic tests to analyze the molecular characteristics of this insect-resistant genetically engineered corn, as well as the diagnostic tests for tracing the SNP in the region of the external insertion to determine the origin and potential diffusion of these maize plants with the AT type or GG type.

## 2. Materials and Methods

### 2.1. Materials and Sampling

We purchased 100 varieties of maize seeds from markets in Jilin and Heilongjiang provinces. Twenty MON810-positive varieties were planted in 12 cm diameter pots and cultured in a greenhouse. The growth conditions were 28 ± 2 °C, 65 ± 10% RH, with a 16 h light/8 h dark photoperiod. After five weeks, we mixed the seedlings derived from 20 varieties with MON810-positive ones for long-read sequencing. A total of 120 strains of MON810 were selected from 20 positive cultivars with MON810 events for AT/GG typing identification (6 individual strains per variety). Four of the twenty positive varieties were selected for BDA enrichment and generation sequencing to detect whether the samples contained low-frequency AT variants. DNA was extracted from powdered seeds and plant tissue (100 ng as starting material) using the Efficient Plant Genome DNA extraction kit (TIANGEN, Beijing, China). DNA concentrations were measured with a Microvolume UV-Vis Spectrophotometer (Thermo Fisher, Waltham, MA, USA).

### 2.2. PacBio-HiFi Sequencing

Positive MON810 samples were planted, and the DNA of leaves from 20 individual plants was extracted and mixed in equal proportions for PacBio-HiFi Sequencing. Target enrichment was used as recommended for the characterization of unknown or partially known GMOs [17]. A bead capture/target enrichment strategy [18] was used to increase the number of relevant sequence reads and increase sequencing mapping accuracy. Using known sequence fragments to capture potential target sequences from measured data according to the principle of sequence base pairing can not only reduce useless sequencing information but also obtain more accurate target fragments, which is called the fishing strategy. Therefore, the full-length sequence, including insert position and direction, can be efficiently obtained by fishing the valid sequence with one known sequence fragment, such as the known *cry1Ab* gene (Fragment 1 in Figure 1A highlighted in green) in this study.

For the analysis of the Pacbio-Hifi sequencing data, we modified a previous protocol [19] as follows. All sequentially obtained reads were aligned pairwise, and all reads that were completely duplicated within other reads under the 99% similarity threshold were removed. The remaining reads were referred to as “best reads” [20]. The best reads were compared in a pairwise manner using Minimap [21] to determine their overlap and build a Hamilton diagram. Mummer [22] was used to compare the reference sequence for targeted capture with all the best reads, to extract the best reads that could be matched to the reference sequence, and then extend both sides to the first fork and determine their extension distance. The contig was reported if the extension distance between the two contig ends from the reference sequence was >20 kb (the default value of 20 kb was sufficient for locating the chromosome on which the insert was located). If it was ≤20 kb before a new fork occurred, recursion was performed on the fork path, and all possible permutations and combinations were listed. If it was ≤20 kb and could not be extended further, the contig was reported.

### 2.3. High-Throughput Sequencing-Based PCR-Free Library

For the next-generation sequencing, a PCR-free library was established using the PCR product from MON810 that was amplified with eight primer pairs. A total of 1.5 μg DNA per sample was used as input material for the DNA library preparation. The sequencing library was generated using NEB Next Ultra II FS DNA PCR-free Library Prep Kit for Illumina (NEB, Ipswich, MA, USA) for each sample. The DNA samples were end-polished, A-tailed, and ligated with the full-length adapter for Illumina sequencing. Subsequently, the DNA products were purified with the AMPure XP system (Beverly, Westlake Village, CA, USA), and their size distribution was analyzed with the Agilent 5400 system (Agilent, Santa Clara, CA, USA) and quantified by quantitative real-time PCR (qPCR). The libraries were pooled and sequenced on Illumina platforms with the PE250 strategy, which takes into account the effective library concentration required. The original fluorescence image files obtained were transformed into short reads (raw data) by base calling, and these short reads were saved as Fastq files [23], which contained both sequence information and sequencing quality information. We used Fastq (version 0.23.1) to perform basic statistics to determine the quality of the raw reads.

The steps of data processing were as follows: (1) Discard paired reads if either one of the pair contains adapter contamination (>10 nucleotides aligned to the adapter, allowing ≤10% mismatches). (2) Discard paired reads if ≥10% of bases were uncertain in either of the pairs. (3) Discard paired reads if the proportion of low-quality (Phred quality < 5) bases was >50% in either of the pairs. Valid sequencing data were then mapped to the reference sequences with the Burrows–Wheeler Aligner (BWA-MEM2) software [24] to obtain the original mapping results stored in BAM format (parameter: mem -t 4 -k 32 -M). Then, the results were emoved duplicates by SAM tools (V1.17) (parameter: rmd up) and Picard (http://broadinstitute.github.io/picard/, accessed on 20 January 2024) [24]. The raw SNP/InDel sets were called using SAM tools with the parameters “-C 50 - m pile up -m2 -F 0.001 -d 1000”. Then, we filtered this set to retain sequences with a depth of the variate position of >1000 and mapping quality of >20.

### 2.4. Design of Diagnostic Assays and Sanger Sequencing

Parallel DNA samples that were identical to those used for high-throughput sequencing were amplified by long fragment overlapping PCR to verify the structure of the MON810 event with the eight pairs of primers shown in Table 1. The optimized reactions were carried out in 20 μL volumes with the following concentrations: 1× Hieff qPCR SYBR Green Master Mix (YEASEN, Shanghai, China), 100 nM of forward and reverse primers (Table 1), and 100 ng template DNA. PCR conditions were as follows: 3 min at 95 °C, 35 cycles of 10 s at 95 °C, 3 min at 68 °C, and 3 min at 72 °C. Amplification products were sequenced to calibrate the high-throughput sequencing data. Multiple amplification products were extracted and purified using the MolPure PCR Purification Kit (YEASEN, Shanghai, China). Sequencing products were run on an Applied Biosystems TM 3730XL (Applied Biosystems, Foster City, CA, USA).

We also carried out PCR using the same conditions as described above with forward and reverse primers at both ends of the MON810 insertion site (Table 1) to determine whether all lines had the same insertion site.

### 2.5. Detection of Variant Type Using Qualitative AS-PCR

Allele-specific PCR (AS-PCR) was carried out with the indicated primers (Table 1). The 20 μL PCR reactions contained 1× Go Taq Green Master mix (Promega, Fitchburg, WI, USA), 100 nM each of forward and reverse primers, and 100 ng template DNA. PCR conditions were as follows: 5 min at 95 °C; 35 cycles of 30 s at 95 °C, 30 s at 62 °C, 30 s at 72 °C; and 3 min at 72 °C. Amplification products were sequenced to confirm the different genotypes. A total of 120 leaves were selected to identify the SNP using the AS-PCR-based construct-specific method.

### 2.6. Detection of Variant Type Using Quantitative AS-PCR-Based Construct-Specific Method

A construct-specific qPCR assay was carried out to quantify the AT genotype. A TaqMan probe (Table 1) was designed to ensure specificity with the same forward and reverse primers used for the construct-specific PCR assay. Four positive varieties of MON810 were selected for quantitative testing to determine whether they contained the AT type. qPCR was carried out in 20 μL volumes containing 1× Probe qPCR Mix (TaKaRa Bio, San Jose, CA, USA), 100 nM each of the forward and reverse primer, 50 nM probe, and 100 ng template DNA. qPCR conditions were as follows: 2 min at 95 °C, 40 cycles of 5 s at 95 °C, and 34 s at 60 °C. The fluorescence signals were collected at 60 °C. qPCR was carried out with an ABI7500 real-time PCR machine (Applied Biosystems, Foster City, CA, USA), and data were analyzed with 7500 Software v2.0 Software (Applied Biosystems). A calibration assay was performed using the event of MON810 for each run to ensure the integrity of the results. The content of the variant type was calculated with the 2^−ΔΔCt^ method [25].

### 2.7. BDA Method to Detect the Low-Frequency AT Variant

Four samples tested by quantitative PCR were used to verify the BDA enrichment method. Sanger sequencing was performed on samples before and after enrichment to verify the enrichment efficiency of the BDA method. BDA enrichment is achieved by enhancing the differential hybridization yield of the preprimer P on the WT template and the variant template, resulting in a difference in amplification efficiency per cycle. This differential amplification yield is compounded, with high enrichment factors produced through multiple PCR cycles. Thus, the rich region of the BDA system corresponds to the nucleotide to which the blocker uniquely binds. The template with sequence variation in the rich region is preferentially amplified. The exact length of this rich region depends on the sequence of this region, but for the experiments shown here, this length ranges from 12 to 30 nucleotides (nt). In this study, however, we chose a simpler approach by adding four unmodified nucleotides to the 3′ end of the blocker that do not pair with the template. For DNA polymerases lacking 3′ to 5′ exonuclease activity, this 3′ termination sequence effectively blocks the expansion of the WT sequence. BDA was carried out in 20 μL volumes containing 1× Master PCR Mix (Promega, Fitchburg, WI, USA), 10 μM each of the forward and reverse primer, 100 Μm blocker, and 50 ng template DNA. The conditions were as follows: 2 min at 95 °C; 45 cycles of 5 s at 95 °C; 30 s at 60 °C; 2 min at 72 °C; 2 min at 72 °C; and 2 min at 10 °C. The samples to be tested need to be controlled without blockers, and the enriched PCR products were sequenced by Sanger sequencing.

## 3. Results

### 3.1. Pacbio-Hifi Sequencing

Twenty of the one hundred market samples were found to be positive for MON810 according to the method reported previously [8]. The Pacbio-HiFi sequencing of the MON810 event was conducted. Pacbio-HiFi sequencing, known *cry1Ab* sequences, was used for the first fishing (①), and seven CCS reads were obtained. On this basis, the obtained sequences were used as known sequences for fishing the 5′ (②) and 3′ (③) flanking sequences. After the Hamilton diagram was constructed, these extended reads could specifically form certain contigs, then, they were reported. Twelve specific and useful CCS reads were obtained after removing duplicates from the above dataset (Appendix A). Fortunately, they were spliced into a contig with a length of 183 kb (Figure 1).

The obtained sequences were spliced and compared with the known sequence from the previous study [17]. The full-length insertion sequence of the MON810 event was obtained, with a length of 37,499 bp, including 9128 bp of 5′ flanking sequence, which was located between 92335503 and 92344630 on chromosome 8. The sequence between the 5′ flanking sequence and the promoter included a retrotransposon that was derived from the region of 55869747–55879326 on chromosome 5 (Figure 1A). Importantly, there was a CaMV35S promoter with a length of 301 bp, an *hsp70* intron sequence from maize with a length of 804 bp, *cry1Ab* with a length of 2457 bp, and a 35 bp linker sequence at the insertion site. The sequence between the linker and 3′ flanking sequence consisted of three fragments that were located in the region of 68416240–68419152 on chromosome 5, 55826390–55831966 on chromosome 5, and the region of 301–958 of the ZM_BFb0017I20 mRNA (Figure 1A). We thus obtained the exact insertion structure and full-length sequence of MON810, which is different from the insertion structure of MON810 as previously reported [17]. The 5′ and 3′ flanking genome sequences were successive to each other on the chromosome, which is scientific and reasonable with the characteristics of an external insertion.

Long-fragment PCR was carried out to amplify the full length of the MON810 insertion event. Subsequent agarose gel electrophoresis showed that the PCR products were bright and clear (Figure 1B). Multiple primers were then used for Sanger sequencing of the PCR products. Multiple sequences obtained from MON810 samples were compared and spliced to obtain the full-length sequence, which was consistent with the sequences obtained with the Pacbio-HiFi sequencing.

### 3.2. Identification of Sequence Diversity at the Insertion Site

The insertion site of the MON810 exogenous gene was a transposon sequence within the maize genome, and there was a deletion of 235 bp (short deletion, D type) of the transposon sequence in the maize genome (Figure 2A). In some MON810 varieties, there was a 370 bp deletion (long deletion, L type). The DL type (D type and L type) was also present in a few plants, and some MON810 plants had no detectable deletion (not detected, N type; Figure 2B). Interestingly, there was diversity in the transposon sequences of different maize lines, such as what occurs in different elite maize lines, including nontransgenic maize, suggesting a sequence diversity that occurs independently of an external insertion. It is worth mentioning that sequence diversity was not detected in insertion sites, which may represent the homozygous insertion of foreign genes. When the insertion site was L-type or D-type, it indicated heterozygous insertion of the foreign gene. DL type indicated nontransgenic corn.

### 3.3. Next-Generation Sequencing

DNA from leaves with the same as Pacbio-Hifi sequencing was amplified based on the full-length amplification system described above, and the products were resequenced to detect whether the SNP was present in MON810 populations. We constructed a PCR-free library, resulting in approximately 1.06 Gb of raw reads and 1.04 Gb of clean reads (98.35%). The percentage of bases with a phred value > 30 as compared with the total bases was 92.08%. The average sequencing depth was up to 1.45 million reads, which was sufficient for an SNP analysis [26]. Interestingly, we found an SNP (AT > GG) (Figure 3A) with high frequency in the *hsp70* intron sequence that linked to the promoter. However, no complementary sequence has been reported in the database of NCBI, suggesting that it may be a newly discovered genotype. Then, the 120 MON810 plants were analyzed to determine their different genotypes, homozygous AT/AT or GG/GG and heterozygous AT/--, GG/--, and AT/GG.

### 3.4. Detection of the Variant Frequency Using Qualitative AS-PCR

AS-PCR was carried out with primers to distinguish between the two genotypes of the SNP of interest (Figure 3A). Two of the primers were designed such that their 3′ end bases were complementary to the mutant (AT) and wild-type (GG) templates. All three primer pairs produced a single bright product, consistent with the expected results (Figure 3A,B). Coincidentally, the mutation site was close to the 5′ flanking sequences, which was convenient for designing an event-specific forward primer. Construct-specific and gene-specific forward primers were also designed (Table 1). For the gene-specific primer, both homozygous and heterozygous samples containing AT were effectively amplified by primers of 810-19041F/810-AT. However, for the primer pair 810-19041F/810-GG, all samples containing AT and/or GG were amplified because we could not distinguish whether the target was derived from the genome sequence. In contrast, event-specific and construct-specific primers were not affected by genomic sequences and could therefore amplify their targets specifically.

We conducted construct-specific PCR using DNA isolated from the 120 leaves (20 varieties × 6 leaves) to identify the genetic characteristics of the AT genotype. The results, which were validated by Sanger sequencing, showed that the number of homozygous plants with the GG type accounted for a large proportion of these leaves, and there was no difference in the ratio of the AT and AT/GG types. There were different proportions of AT and GG genotypes in each variety. It can be inferred that the parents of these 120 samples from 20 varieties may belong to three main genotypes, such as AT (homozygous AT/AT and heterozygous AT/--), GG (homozygous GG/GG and heterozygous GG/--), and AT/GG, respectively. In addition, the similar number of homozygous AT and heterozygous AT/GG types suggest that the occurrence of AT may be related to the AT/GG genotype at the population level.

### 3.5. Detection of the AT Variant Frequency Using Quantitative AS-PCR

The standard curve established for the MON810 event was y = −3.5072x + 40.1827, with the value of R^2^ being 0.9987. And for the AT type, the standard curve was y = −3.5890x + 39.8607, with the value of R^2^ being 0.9920. The standard curve meets the requirement of quantitative PCR (−3.6 ≤ slope ≤ −3.1; R^2^ ≥ 0.98). To clarify the genetic evolution of the AT genotype, we carried out construct-specific qPCR on the four positive MON810. There were two samples containing the AT type (14.8% and 98%) using qPCR, suggesting that the AT mutation had arisen early in China. Because of the high percentage of the AT type in early samples, it is likely that the AT mutation was present in the MON810 population at a high level and will continue to exist within this population.

### 3.6. BDA Enrichment to Detect Low Variant Allele Frequency (VAF) of the AT Type

We then conducted a BDA assay in conjunction with the construct-specific method using the same samples selected from Section 3.5, described to look for low VAF associated with the AT type (Figure 4A,B). The reverse blocker was selected because it inhibited the amplification of the wild-type sequences efficiently compared with the forward blocker (Figure 4C).

This was a highly efficient method for enriching low VAF on a scale of 0.1–100%, indicating high stability (Figure 4D). Some materials were tested using the BDA system, of which two of the four showed a very faint amount of the AT type before adding the blocker but showed an obvious peak or a stronger signal peak for the AT type when the reverse blocker was added, suggesting a highly efficient enrichment of low VAF. There was no AT signal in S01, which probably corresponded to the GG type (i.e., wild type). S02 and S03 had an obvious AT signal after the blocker; however, it was GG type using Sanger sequencing before adding the blocker in S02. It is worth mentioning that the enrichment levels of the two are different, perhaps because the content of S02 was low, which is consistent with the results of quantitative AS-PCR. And the S04 was not changed, because it was AT type before adding the blocker (Figure 4E,F). Importantly, there were samples that we analyzed that had an AT signal in the presence of the blocker. Thus, this method is better able to detect low-frequency variant alleles.

## 4. Discussion

The application of genetically modified crops has brought enormous economic, social, and environmental benefits [27]. Here, we used a Pacbio-HiFi sequencing base fishing strategy and Sanger sequencing to reconstruct and characterize a genetically modified maize variety grown in China for which the knowledge of the insertion event was incomplete. Further testing led to the identification of the entire landscape of the insert of MON810. The complete insertion structure of MON810 was obtained for the first time through third-generation long-read length sequencing, and the flanking genomic sequence was consistent with the genome at the insertion site. Thus, a truly complete full-length insertion sequence was obtained, which resulted in a revision of previously reported data on the exogenous insertion structure and full-length sequence of MON810. On this basis, we found an SNP (AT > GG) located in the *hsp70* intron sequence and then used different detection methods to construct and detect the AT type in simple and complex samples.

The PacBio sequencing technology was produced using the cyclic consensus sequencing (CCS) mode on the PacBio long-read system [19]. PacBio-HiFi sequencing can not only identify homozygotes and heterozygotes but also span complex structures, such as tandem repeats. The true long reads of HiFi sequencing have proven useful for the characterization of GMOs, especially for events that are relatively complex or include large insertions, repeat sequences, and/or multiple copies of insertions, such as Zhonghuang 6106 (ZH10-6), which was an herbicide-tolerant genetically modified soybean with an inverted repeat structure and genome fragment in the region of the insertion (unpublished results). Importantly, we were able to obtain an accurate location of the MON810 insertion thanks to the long read length (>9 kb). Previous studies [17] have obtained incorrect chromosome localization information because the sequencing read length was insufficient. Moreover, the chromosome localization information of the MON810 insert obtained in this study was consistent with PCR verification, indicating that this is the complete insert structure.

In the first round of our analysis, we used a fishing strategy to obtain the sequence using the known *cry1Ab* reading frame as the “bait”. Fortunately, seven long reads were obtained that spanned the *cry1Ab* reading frame area and extended some distance into the 5′ flanking and 3′ flanking regions. One end of the seven long reads extended into the genome sequence and beyond the position previously reported as the 5′ flanking sequence [17], whereas the 3′ flanking extended into the region closer to the 5′ flanking of the genome, indicating that it was close to complete insertion information. The valid sequences obtained in the first round were used to obtain a more accurate insertion position in the second round with the fishing method. Finally, we obtained the precise full-length sequence of the MON810 event, including five fragments with an incomplete *cry1Ab* reading frame and four genome fragments (Figure 1A,B). It was an accurate full-length sequence of the MON810 event, because the two ends of the insertion sequence were close to each other, which is consistent with the characteristics of Agrobacterium transformation, whereas the two ends of the insertion sequence were far apart from each other, which was not logical in the MON810 event in a previous report [28,29]. Therefore, we should obtain a complete and logical flanking sequence and genome sequence when locating the location of the event. If the two sides of the genome sequence were obtained, the side sequence was directly judged to be a flanking sequence, resulting in ignoring the information that may represent the true insertion location. Obviously, basic logic like this can be used in the process of analyzing the characterization of events to improve the accuracy of the results for the insertion information. In addition, according to the 12 long reads of HiFi sequencing result, seven of them covered the junction of 35S/*hsp70*, and the AT genotype was detected in three of the seven reads, indicating that HiFi sequencing could detect a few base variants. The result was also confirmed by NGS data based on PCR-free sequencing.

Therefore, this study not only obtained the correct full-length insertion structure of MON810 but also found the SNP located in the regulatory element region and established some methods for this SNP site suitable for different application scenarios. Direct Sanger sequencing using PCR amplification products can obtain high-precision sequence information, which was one of the precise methods. However, considering its low flux, this study had appropriately increased the efficiency of first-generation sequencing, which can obtain the sequence information of four samples in a single singer sequencing. If there were any changes in the four sequences, it can be observed through the peak map to further determine whether there was variation directly. For AS-PCR, this study conducted gene-specific, construct-specific, and event-specific methods to determine whether there was an “AT” variant type compared with the original sequences “GG” in the population of MON810 events. The AT mutations occurred in the *hsp70* sequence, so gene-specific PCR was applicable to all events containing the *hsp70* gene. However, construct-specific PCR has a higher specificity and is applicable to distinguish different constructs containing the same elements. In this study, it can be applied to the junction regions containing 35S promoters and *hsp70*, such as some events that have been commercialized, except for MON810, such as MON87411, MON87427, DBN9936, DBN9501, and NK603. The construct-specific detection method established can also detect whether there is an AT type in these events and could help clarify whether AT variation was a specific event occurring in the events or a common phenomenon in different events. On this basis, the event-specific method was vital to detect the AT type and determine whether it was the event MON810. It can even be used to detect the AT variation in MON810 in complex mixed samples composed of different events, and it is convenient for researchers or monitoring agencies because there is no need to carry out too much testing. Compared with other molecular diagnostic technologies used for the detection and quantitation of rare alleles from clinical samples, BDA was unique in simultaneously providing good mutation sensitivity, high mutation multiplexing, fast turnaround, and low reagent and instrument cost. Furthermore, in contrast to many proof-of-concept works in the academic literature showing high mutation sensitivity against one or a few mutations, this study established a BDA method to detect low VAF efficiently by blocking the WT sequences and enriching the variant sequences [30]. The method was established based on a construct-specific method, which was more accurate than gene-specific methods because it linked the p35S and the *hsp70* regions that could distinguish whether AT type was present in events containing the same construct. In addition, the blocker overlapped the primer by 10 bp, and four “A” bases were added to the 3′ terminal of the blocker, which could effectively block the amplification of WT-type sequences and amplify low-frequency VAF. Importantly, BDA combined with Sanger sequencing can intuitively determine whether there was VAF as low as 0.1% in the samples to be tested, which was impossible for conventional methods. Thus, the method was an efficient and simple method to enrich low-frequency minority base variation.

The origin analysis of maize seed purchased in China found that the frequency of the AT type was slightly high, but the cause of its occurrence was not clear. We suspect one of two likely explanations: (1) The mutation occurred during the breeding process abroad, persisted at a moderate level for a long time, and entered the country with imported maize seeds. (2) Alternatively, the AT mutation occurred in China, was introduced artificially or accidentally during the early stage of breeding, and persisted in a high proportion, such that our study detected a high frequency of occurrences in several domestic samples. The quantitative detection of the AT type based on the construct-specific method can not only detect the AT type but also quantify the AT content in complex samples, which will be helpful for the monitoring of corn samples for AT type. The occurrence of a high-frequency SNP in genetically engineered insertions has rarely been reported and is an interesting phenomenon. Specifically, whether the AT mutation has a regulatory effect on the 35S promoter to further affect the expression of the *cry1Ab* insecticidal gene is worth exploring and may be relevant for pest control. In addition, the sequence change from GG to AT results in a corresponding change in the complementary strand from CC to TA. Whether this will lead to a change in cytosine-related methylation and subsequent epigenetic effects is unknown. The subsequent exploration of the expression of *cry1Ab* in homozygous and heterozygous genotypes with respect to the AT type should be carried out.

## Figures and Tables

**Figure 1 plants-13-02276-f001:**
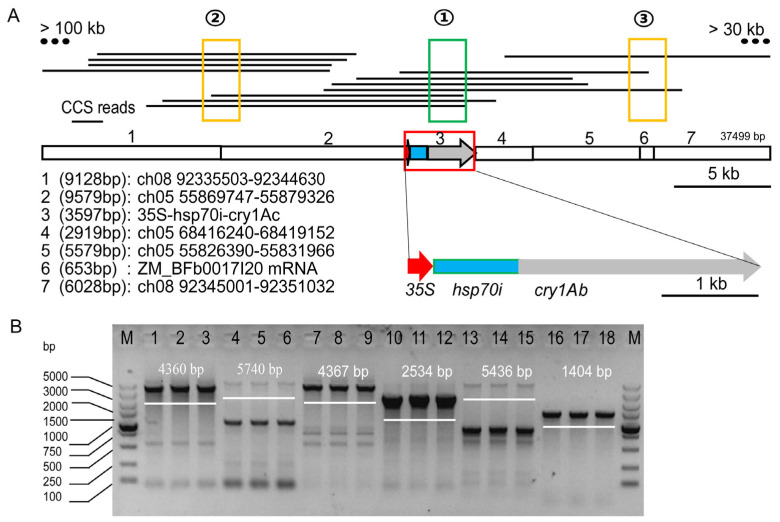
The overall description of sequences inserted in association with the MON810 event: (**A**) The full-length inserted sequence and orientation of the inserted elements. Location information for each of the 12 efficient CCS reads with respect to the external insertion sequences associated with the MON810 event. (**B**) Segmentally amplified gel electrophoresis map of insertion sequences for the MON810 event. The length of the fragments was 5015 bp, 5107 bp, 4935 bp, 4929 bp, 5169 bp, 5039 bp, 5201 bp, and 4930 bp.

**Figure 2 plants-13-02276-f002:**
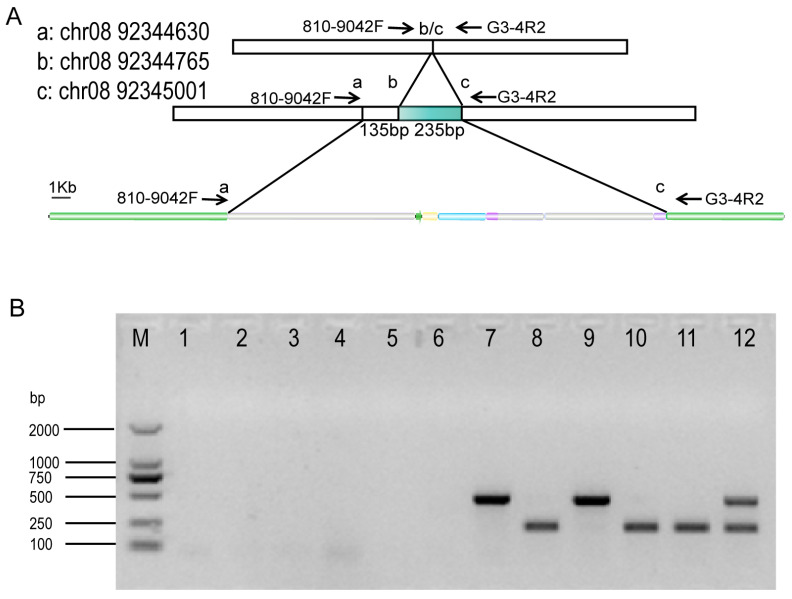
Genome sequence diversity diagram of MON810: (**A**) Genome sequence diversity and primer design of MON810 insertion site. There were some deletions of 235 and/or 370 bp (135 bp plus 235 bp), respectively. (**B**) Diversity map of insertion sites in maize genome. Lane 1–6 indicate N type. Lane 7, 9 indicate L type. Lane 8, 10, 11 indicate D type. Lane 12 indicates DL type.

**Figure 3 plants-13-02276-f003:**
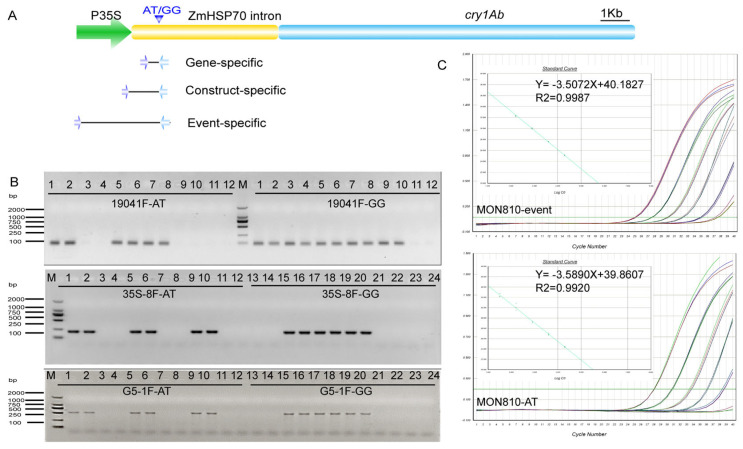
The specific-method-based AS-PCR for AT type: (**A**) There were primers designed for gene-specific, construct-specific, and event-specific detection methods of the AT type. (**B**) Identification of the AT and GG types. AT primers can detect not only homozygous AT types but also heterozygous AT/GG types, and similarly, GG primers can detect not only heterozygous AT/GG types but also homozygous GG types. (**C**) Quantitative standard curves of MON810 events and AT genotypes. DNA samples of AT-type homozygous MON810 were successively diluted to 100 ng, 10 ng, 5 ng, 1 ng, 0.1 ng, and 0.01 ng as the DNA template of the standard curve. The MON810 content was determined, and then the frequency of the AT type was determined relative to the MON810 content.

**Figure 4 plants-13-02276-f004:**
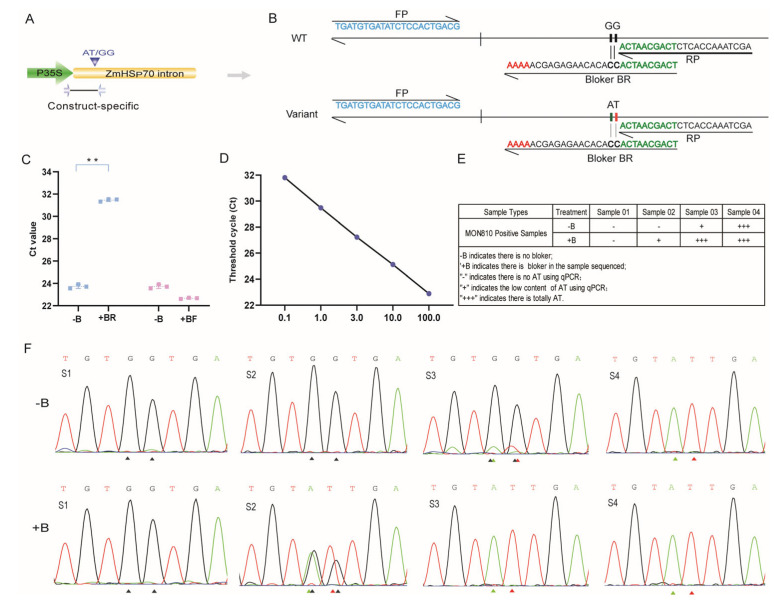
BDA analysis of the AT type variant. (**A**,**B**) Primers and blocker designs based on build-specific BDA methods to block GG amplification and amplify the AT sequence. (**C**) Comparison of blocking efficiency of the forward (BF) and reverse blocker (BR) compared with amplification in the absence of the blocker. ** means significant difference (*t*-test, *p* < 0.01). (**D**) Amplification effect of the BDA method on different enrichment targets. The abscissa represents the different contents of the AT-type sample; the contents were 0.1%, 1%, 3%, 10%, and 100%, respectively. (**E**,**F**) Sanger sequencing of the products enriched by BDA can determine whether the AT type was present in the mixture materials. The AT type was greatly enriched and amplified, which was demonstrated by Sanger sequencing peak maps of the amplified product from the samples. “FP” means forward primer, and “RP” means reverse primer. The red triangle symbol represents thymine (T), the green triangle symbol represents adenine (A), and the black triangle symbol represents guanine (G).

**Table 1 plants-13-02276-t001:** Primers used in this study.

Assay	Target	Primer	Sequence (5′–3′)	Product (bp)
MON810 full length	The full length of MON810	MON810_1F	ATGACCAGGGGTACGTTCGATA	5015
		MON810_1R	CGTTGAGCAATCAAAGCGTGAG	
		MON810_2F	TCTGCGACTTCTTCAGCTGTTC	5107
		MON810_2R	AGTCACTAGGTGGTTGGAGTGA	
		MON810_3F	GAGCAACGTCTACTTCGCGTAA	4935
		MON810_3R	GACTACACAATCACTTGGCCGT	
		MON810_4F	TACGGAGTCCAAAAGTTGCCG	4929
		MON810_4R	TTTCGGGCGAAGGTTATGAAGG	
		MON810_5F	CGGCTTCTGAAGGTCCTCAAAA	5169
		MON810_5R	ATACTTCCCGGCGGATACTGAT	
		MON810_6F	AAGTCAGACGAGACCCTCCAAT	5039
		MON810_6R	GATTCAATCCCAGGCGTTAGCT	
		MON810_7F	TGTTGAGGACCGCTCTTTCAAG	5201
		MON810_7R	CCCTGCGATAAAGTTAGCCCAT	
		MON810_8F	AGCTAAGGGGGTTAAACAACTTGT	4930
		MON810_8R	AGGTCTGAGTTGGCGTGAGATA	
Insertion DL	Detection for Insertion site	G3-4R2	CTTTACCACAAGAGATAAGG	235/370
		810-9042F	AGAGACGAATAAGCAAGTTAGC	
Gene specific	SNP for AT type	810-19041F	CTTCGGTACGCGCTCACTCC	GG 88/AT 92
Construct specific		35S-8F	TAAGGGATGACGCACAATCC	GG 200/AT 204
Event specific		G5-1F	TATGTCCTTCATAACCTTCG	GG 513/AT 517
AT type		810-AT	GCTAAACCACTCTCAGCAATCAAT	/
GG type		810-GG	AACCACTCTCAGCAATCACC	
BDA	BDA method	810-18907F	TGATGTGATATCTCCACTGACG	/
		810-19133R	AGCTAAACCACTCTCAGCAATCA	227
		810B-19120R	TCAGCAATCACCACACAAGAGAGCAAAAA	214

## Data Availability

Data related to this study can be found in the article or Appendix A; further inquiries can be directed to the corresponding author.

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
