# Peer review of "An Integrated Strategy for Analyzing the Complete Complex Integrated Structure of Maize MON810 and Identification of an SNP in External Insertion Sequences"

_plants, 2024, doi:10.3390/plants13162276_

Round 1

Reviewer 1 Report

Comments and Suggestions for Authors

The article “An Integrated Strategy for Analyzing the Complete Complex 2 Integrated Structure of Maize MON810 and Identification of a 3 SNP in External Insertion Sequences” deals with the identification of the integration event MON810 and of a SNP in the HSP70 intron that likely originated in China and that is the first reported variant identified in the transgenic line.

The article can fit with the scope of “Plants” but it can not be published in the present version.

Main problems:

1) the article is not easy to be followed, it needs to be deeply reorganised to improve comprehensibility and easiness of reading.

2) a deep revision of the English style by a mother tongue is required to simplify the different paragraphs and improve reading.

3) at least half of the manuscript is based on the characterisation of a SNP that was found in Chinese varieties including the MON810 event. Why is it important? Truly, can the presence of that SNP justifies so big space in analysing and describing it? Why is the presence of the SNP more than being a curiosity?

Subsequently some of the points that can be improved

1) introduction: lines between 78 and 85 can be removed, it is not necessary to report the functioning of Pac-bio

2) material and methods:

a) paragraph 2.1 materials and sampling, please provide more details (eventually as supplementary tables) concerning the varieties that were sampled, evidencing also the varieties that were subsequently found to be GM, and the reference samples purchased from EU.

b) paragraph 2.2, it is written “……..Therefore, the full-length sequence, including insert position and direction, can be efficiently obtained by fishing the valid sequence with four known sequence fragments, such as W05-1, W05-1 plus cry1Ab, W03-1 149 plus cry1Ab, and W03-1 in this study.” Please, provide more details about these fragments.

c) paragraph 2.4. They report a table with the primers that were used in the study. The table is fine, eventually it can be moved in supplementary information, I suggest to insert in the manuscript a figure reporting the different amplified traits of the region of interest by each primer pair.

d) paragraph 2.6. I do not think the title is corresponding to the matter of the paragraph.

Results

1) paragraph 3.1. Pacbio-Hifi Sequencing. They write that “Sixteen of 100 market samples were found to be positive for MON810 according to the method reported previously (Holck et al., 2002). The Pacbio-Hifi sequencing of 16 positive MON810 samples were conducted”. Please, clarify which are the positive samples. Did the sequencing of the samples provide exactly the same results in all of them?

Figure 1 is not necessary, it can be removed.

Paragraph 3.1. it is written, lines 58-61 “Furthermore, of these 12 contigs (Fig. 2A, B), four contained only the W5-01 fragment, two contained both W5-01 and cry1Ab fragments, four contained only cry1Ab fragments, one contained both the W3-01 and cry1Ab fragments, and one contained only the W3-01 fragment.” I suggest a specific figure (eventually in supplementary information) that can be useful to explain better the different contigs and the included regions.

Figure 2.

Figure 2 A and text, so, are you stating that the integration, that involves a big rearrangement of the genome, is not in chromosome 5 but in chromosome 8? This is what it seems from figure 2A, is it correct? Please, clarify better

Figure 2 B is not so informative in absence of a figure reporting where the different contigs are located.

Figure 2 D is not described in the text

Figure 2 F it is written “vartity” is it correct?

Paragraph 3.1. it is written, lines 88-92, “We had thus obtained the exact insertion structure and full-length sequence of MON810, which is different from the insertion structure of MON810 as previously reported (Zhang et al., 2022). The 5’ and 3’ flanking genome sequences were successive to each other on the chromosome, which is scientific and reasonable with the characteristics of an external insertion.” To say so, they should consider also the material that was analysed in the previous paper. Did they do this? Did the sequence also the same material?

Paragraph 3.3 Identification of Sequence Diversity at the Insertion Site

Please, associate the different deletions with the corresponding variety (name of the variety-kind of deletion).

Discussion

More than half of the discussion is based on the identified SNP. But, again, is this so important to justify a so big amount of description? I suggest to shorten this part.

Comments on the Quality of English Language

The English style must be improved. A revision from a mother tongue is necessary.

Author Response

Dear reviewers,

Thanks very much for taking your time to review this manuscript. I really appreciate all your comments and suggestions on the manuscript include some details, etc. Please find my itemized responses in below and my revisions/corrections in the re-submitted files (highlighted in green). Thanks again!

The article “An Integrated Strategy for Analyzing the Complete Complex 2 Integrated Structure of Maize MON810 and Identification of a 3 SNP in External Insertion Sequences” deals with the identification of the integration event MON810 and of a SNP in the HSP70 intron that likely originated in China and that is the first reported variant identified in the transgenic line.

The article can fit with the scope of “Plants” but it can not be published in the present version.

Main problems:

Question 1: the article is not easy to be followed, it needs to be deeply reorganised to improve comprehensibility and easiness of reading.

Reply to question 1: Thank you for the your suggestion. I have revised the manuscript with the help of my native English speaking colleague. I hope it will improve the quality of the manuscript.

Question 2: a deep revision of the English style by a mother tongue is required to simplify the different paragraphs and improve reading.

Reply to question 2: We are grateful for the suggestion about the writing. I have revised the manuscript with the help of my native English speaking colleague. I hope it will improve the quality of the manuscript. In addition, i have shorten some paragraphs to improve reading.

Question 3: at least half of the manuscript is based on the characterisation of a SNP that was found in Chinese varieties including the MON810 event. Why is it important? Truly, can the presence of that SNP justifies so big space in analysing and describing it? Why is the presence of the SNP more than being a curiosity?

Reply to question 3: We are grateful for your suggestion about the whole manuscript. The MON810 was an important event because of the unauthorized cultivation of identity in China. However, it was detected in some market samples. What’s more, the insertion location was wrongly reported, and we reported the right insertion location and full sequences by pacbio-HiFi sequencing. The SNP located in insertion sequence especially in regulatory region is interesting and maybe important for the expression of cry1Ab gene. maybe it was founded first time., so, I discussed the SNP in some extent. According to your suggestion, I simplify the discussion of the SNP. Please refer to line 249-304, page 14 and 15 of 17.

Subsequently some of the points that can be improved

Question 4: introduction: lines between 78 and 85 can be removed, it is not necessary to report the functioning of Pac-bio

Reply to question 4: We are grateful for the suggestion. I have simplified the description of the HiFi sequencing, please refer to line 78-82, page 2 of 17.

2) material and methods:

Question 5: a) paragraph 2.1 materials and sampling, please provide more details (eventually as supplementary tables) concerning the varieties that were sampled, evidencing also the varieties that were subsequently found to be GM, and the reference samples purchased from EU.

Reply to question 5: We are grateful for the suggestion. We have corrected the material information. Finally, we do not use the giving materials and purchased standard substances from EU. please refer to line 123-129, page 3 of 17.

Question 6: b) paragraph 2.2, it is written “……..Therefore, the full-length sequence, including insert position and direction, can be efficiently obtained by fishing the valid sequence with four known sequence fragments, such as W05-1, W05-1 plus cry1Ab, W03-1 149 plus cry1Ab, and W03-1 in this study.” Please, provide more details about these fragments.

Reply to question 6: We are grateful for the suggestion. We have revised the information about sequencing to make it easier to understand. “Using known sequence fragments to capture potential target sequences from measured data according to the principle of sequence base pairing can not only reduce useless sequencing information, but also obtain more accurate target fragments, which is called fishing strategy. Therefore, the full-length sequence, including insert position and direction, can be efficiently obtained by fishing the valid sequence with one known sequence fragments, such as known cry1Ab gene (Fragment 1 in figure 1A highlighted in green) in this study. ” line 141-147, page 3 of 17.

Question 7: c) paragraph 2.4. They report a table with the primers that were used in the study. The table is fine, eventually it can be moved in supplementary information, I suggest to insert in the manuscript a figure reporting the different amplified traits of the region of interest by each primer pair.

Reply to question 7: We are grateful for the suggestion. I added the description of targets of some PCR in table1. page 6 of 17.

Question 8: d) paragraph 2.6. I do not think the title is corresponding to the matter of the paragraph.

Reply to question 8: We are grateful for the suggestion. I changed the title according to your suggestion. “Detection of Variant Type Using Quantitative AS-PCR based Construct-specific Method” . line 12, page 8 of 17.

Results

Question 9: paragraph 3.1. Pacbio-Hifi Sequencing. They write that “Sixteen of 100 market samples were found to be positive for MON810 according to the method reported previously (Holck et al., 2002). The Pacbio-Hifi sequencing of 16 positive MON810 samples were conducted”. Please, clarify which are the positive samples. Did the sequencing of the samples provide exactly the same results in all of them?

Reply to question 9: We are grateful for the suggestion. I have revised the result according to our assay record. There were 20 positive varieties containing MON810 event. “20 varieties with MON810 positive were planted in 12 cm diameter pots and cultured in a greenhouse. After five weeks, mixed the seedlings derived from 20 varieties with MON810 positive for long-read sequencing (one leaves in each variety)”.

Question 10: Figure 1 is not necessary, it can be removed.

Reply to question 10: We are grateful for the suggestion. I have removed the Figure 1 according to your suggestion.

Question 11: Paragraph 3.1. it is written, lines 58-61 “Furthermore, of these 12 contigs (Fig. 2A, B), four contained only the W5-01 fragment, two contained both W5-01 and cry1Ab fragments, four contained only cry1Ab fragments, one contained both the W3-01 and cry1Ab fragments, and one contained only the W3-01 fragment.” I suggest a specific figure (eventually in supplementary information) that can be useful to explain better the different contigs and the included regions.

Reply to question 11: We are grateful for the suggestion. I changed the figure 1A and have revised the description of the result. For the figure 1A, using known sequence fragments to capture potential target sequences from measured data according to the principle of sequence base pairing can not only reduce useless sequencing information, but also obtain more accurate target fragments, which is called fishing strategy. Therefore, the full-length sequence, including insert position and direction, can be efficiently obtained by fishing the valid sequence with one known sequence fragments, such as known cry1Ab gene. Location information for each of the 12 efficient CCS reads with respect to the external insertion sequences associated with the MON810 event was highlighted in the figure 1A.

Figure 2.

Question 12: Figure 2 A and text, so, are you stating that the integration, that involves a big rearrangement of the genome, is not in chromosome 5 but in chromosome 8? This is what it seems from figure 2A, is it correct? Please, clarify better

Reply to question 12: We are grateful for the suggestion. The chromosomal location of the exogenous insertion sequence should be on chromosome 8, and the exogenous sequence consists of several fragments, including the recombination fragment from chromosome 5. I have revised figures, please refer to figure1A and the description of 1A. page 9 of 17.

Question 13: Figure 2 B is not so informative in absence of a figure reporting where the different contigs are located.

Reply to question 13: We are grateful for the suggestion. I have revised figures, please refer to figure1A and the description of 1A. there were more details about the different contigs are located. page 9 of 17.

Question 14: Figure 2 D is not described in the text

Reply to question 14: We are grateful for the suggestion. I have removed the figure and the description in the manuscript.

Question 15: Figure 2 F it is written “vartity” is it correct?

Reply to question 15: We are grateful for the suggestion. I have revised the word “variety” and removed the figure according to your suggestion like the question 14. 

Question 16: Paragraph 3.1. it is written, lines 88-92, “We had thus obtained the exact insertion structure and full-length sequence of MON810, which is different from the insertion structure of MON810 as previously reported (Zhang et al., 2022). The 5’ and 3’ flanking genome sequences were successive to each other on the chromosome, which is scientific and reasonable with the characteristics of an external insertion.” To say so, they should consider also the material that was analysed in the previous paper. Did they do this? Did the sequence also the same material?

Reply to question 16: We are grateful for the suggestion. I have revised the  description of this result. We were sure that the result was consistent with previous studies, but with more complete sequences obtained. The previously reported sequences are part of the reported sequences in this study.

Question 17: Paragraph 3.3 Identification of Sequence Diversity at the Insertion Site. Please, associate the different deletions with the corresponding variety (name of the variety-kind of deletion).

 Reply to question 17: We are grateful for the suggestion. I removed the result of the different deletions because of the absence of corresponding varieties. And I removed the the figure.

Discussion

Question 18: More than half of the discussion is based on the identified SNP. But, again, is this so important to justify a so big amount of description? I suggest to shorten this part.

Reply to question 18: We are grateful for the suggestion. I have shorten the  description of the SNP in the part of result. Please refer to lin 247-252, page 14 of 17, and line 294-304, page 15 of 17.

Reviewer 2 Report

Comments and Suggestions for Authors

The manuscript presents a comprehensive genetic characterization of MON810 corn variants using advanced sequencing and molecular techniques. The study utilizes PacBio-HiFi sequencing, Sanger sequencing, BDA enrichment, qPCR, and AS-PCR to investigate the genetic diversity and evolution of the AT variant in MON810 corn. Key findings include the identification of SNP variants, detailed insertion site analysis, and the detection of variant allele frequencies using BDA enrichment. The study provides insights into the genetic makeup of MON810, correcting previous genomic data discrepancies and highlighting implications for regulatory practices in agricultural biotechnology.

Detailed comments:

1.     Consider providing more detailed explanations in the methods section regarding specific software versions used for data analysis (e.g., BWA, SAMtools). This would enhance reproducibility and allow readers to assess the robustness of the bioinformatic analyses.

2.     Clarify the rationale for selecting specific primer pairs and assays in the methods section, particularly for PCR and AS-PCR experiments. Justify why these assays were chosen over alternative methods and discuss any limitations associated with their application.

Author Response

Dear reviewers,

Thank you very much for taking your time to review this manuscript. I really appreciate all your comments and suggestions on the manuscript including figures, references and writing details, etc. I believe that these changes are conducive to the smooth publication of the manuscript. Please find my itemized responses in below and my revisions/corrections in the re-submitted files (highlighted in yellow). Thanks again!

The manuscript presents a comprehensive genetic characterization of MON810 corn variants using advanced sequencing and molecular techniques. The study utilizes PacBio-HiFi sequencing, Sanger sequencing, BDA enrichment, qPCR, and AS-PCR to investigate the genetic diversity and evolution of the AT variant in MON810 corn. Key findings include the identification of SNP variants, detailed insertion site analysis, and the detection of variant allele frequencies using BDA enrichment. The study provides insights into the genetic makeup of MON810, correcting previous genomic data discrepancies and highlighting implications for regulatory practices in agricultural biotechnology.

Detailed comments:

Question 1: Consider providing more detailed explanations in the methods section regarding specific software versions used for data analysis (e.g., BWA, SAMtools). This would enhance reproducibility and allow readers to assess the robustness of the bioinformatic analyses.

Reply to question 1: We are grateful for the suggestion. I have revised the description of specific software versions used for data analysis, such as, BWA-MEM, SAM tools (v1.17). Please refer to manuscript highlighted in yellow, line 188 and 190, page 4 of 17.

Question 2: Clarify the rationale for selecting specific primer pairs and assays in the methods section, particularly for PCR and AS-PCR experiments. Justify why these assays were chosen over alternative methods and discuss any limitations associated with their application. 

Reply to question 2: We are grateful for the suggestion on the method. We have revised the description of AS-PCR methods in the Methods section, including the way materials and results. The advantages and application scenarios of the selection method are included in the discussion section. For more details, please refer to the the manuscript highlighted in yellow, line 4-17, page 8 of 17.